# Influence Maximization with $\varepsilon$-Almost Submodular Threshold Functions

**Qiang Li**[*†], **Wei Chen**[‡], **Xiaoming Sun**[*†], **Jialin Zhang**[*†]
[*]CAS Key Lab of Network Data Science and Technology,
Institute of Computing Technology, Chinese Academy of Sciences
[†]University of Chinese Academy of Sciences
[‡]Microsoft Research
{liqiang01,sunxiaoming,zhangjialin}@ict.ac.cn
weic@microsoft.com

## Abstract

Influence maximization is the problem of selecting $k$ nodes in a social network to maximize their influence spread. The problem has been extensively studied but most works focus on the submodular influence diffusion models. In this paper, motivated by empirical evidences, we explore influence maximization in the non-submodular regime. In particular, we study the general threshold model in which a fraction of nodes have non-submodular threshold functions, but their threshold functions are closely upper- and lower-bounded by some submodular functions (we call them $\varepsilon$-almost submodular). We first show a strong hardness result: there is no $1/n^{\gamma/c}$ approximation for influence maximization (unless P = NP) for all networks with up to $n^{\gamma}$ $\varepsilon$-almost submodular nodes, where $\gamma$ is in (0,1) and $c$ is a parameter depending on $\varepsilon$. This indicates that influence maximization is still hard to approximate even though threshold functions are close to submodular. We then provide $(1-\varepsilon)^{\ell}(1-1/e)$ approximation algorithms when the number of $\varepsilon$-almost submodular nodes is $\ell$. Finally, we conduct experiments on a number of real-world datasets, and the results demonstrate that our approximation algorithms outperform other baseline algorithms.

## 1 Introduction

Influence maximization, proposed by Kempe, Kleinberg, and Tardos [1], considers the problem of selecting $k$ seed nodes in a social network that maximizes the spread of influence under pre-defined diffusion model. This problem has many applications including viral marketing [2, 3], media advertising [4] and rumors spreading [5] etc., and many aspects of the problem has been extensively studied.

Most existing algorithms for influence maximization, typically under the independent cascade (IC) model and the linear threshold (LT) model [1], utilize the submodularity of the influence spread as a set function on the set of seed nodes, because it permits a $(1-1/e)$-approximation solution by the greedy scheme [1, 6, 7], following the foundational work on submodular function maximization [8]. One important result concerning submodularity in the influence model is by Mossel and Roch [9], who prove that in the general threshold model, the global influence spread function is submodular when all local threshold functions at all nodes are submodular. This result implies that "local" submodularity ensures the submodularity of "global" influence spread.

Although influence maximization under submodular diffusion models is dominant in the research literature, in real networks, non-submodularity of influence spread function has been observed. Backstrom et al. [10] study the communities of two networks LiveJournal and DBLP and draw

pictures of the impulse that a person joins a community against the number of his friends already in this community. The curve is concave overall, except that a drop is observed in first two nodes. Yang et al. [11] track emotion contagion under *Flickr* and find that the probability that an individual becomes happy is superlinear to the number of his happy friends with higher PageRank scores. These are all instances of non-submodular influence spread functions.

Influence maximization under many non-submodular diffusion models are proved to be hard to approximate. For example, in the diffusions of rumors, innovations, or riot behaviors, the individual in a social network is activated only when the number of her neighbors already adopting the behavior exceeds her threshold. It has been shown that the influence maximization problem based on this fixed threshold model cannot be approximated within a ratio of $n^{1-\varepsilon}$ for any $\varepsilon > 0$ [1]. Meanwhile Chen [12] proves that the seed minimization problem, to activate the whole network with minimum size of seed set, is also inapproximable, in particular, within a ratio of $O(2^{\log^{1-\varepsilon} n})$.

In this paper we give the first attempt on the influence maximization under the non-submodular diffusion models. We study the general threshold model in which a fraction of nodes have non-submodular threshold functions, but their threshold functions are closely upper- and lower-bounded by some submodular functions (we call them $\varepsilon$-almost submodular). Such a model bears conceptual similarity to the empirical finding in [10, 11]: both studies show that the influence curve is only slightly non-concave, and Yang et al. [11] further shows that different roles have different curves — some are submodular while others are not, and ordinary users usually have behaviors close to submodular while opinion leaders may not. We first show a strong hardness result: there is no $1/n^{\frac{\gamma}{c}}$ approximation for influence maximization (unless P = NP) for all networks with up to $n^\gamma$ $\varepsilon$-almost submodular nodes, where $\gamma$ is in $(0, 1)$ and $c$ is a parameter depending on $\varepsilon$. On the other hand, we propose constant approximation algorithms for networks where the number of $\varepsilon$-almost submodular nodes is a small constant. The positive results imply that non-submodular problem can be partly solved as long as there are only a few non-submodular nodes and the threshold function is not far away from submodularity. Finally, we conduct experiments on real datasets to empirically verify our algorithms. Empirical results on real datasets show that our approximation algorithms outperform other baseline algorithms.

**Related Work.** Influence maximization has been well studied over the past years [13, 6, 7, 14, 15]. In particular, Leskovec et al. [6] propose a *lazy-forward* optimization that avoids unnecessary computation of expected size. Chen et al. [7, 14] propose scalable heuristic algorithms that handle network of million edges. Based on the technique of *Reverse Reachable Set*, Borgs et al. [16] reduce the running time of greedy algorithms to near-linear under the IC model [1]. Tang et al. [17] implement the near-linear algorithm and process *Twitter* network with million edges. Subsequently, Tang et al. [18] and Nguyen et al. [19] further improve the efficiency of algorithms. These works all utilize the submodularity to accelerate approximation algorithms.

Seed minimization, as the dual problem of influence maximization, is to find a small seed set such that expected influence coverage exceeds a desired threshold. Chen [12] provide some strong negative results on seed minimization problem under fixed threshold model, which is a special case of general threshold model where its threshold function has breaking points. Goyal et al. [20] propose a greedy algorithm with a bicriteria approximation. Recently, Zhang et al. [21] study the probabilistic variant of seed minimization problem.

Due to the limitation of independent cascade and linear threshold model, *general threshold model* has been proposed [1, 9]. Not much is known about the general threshold model, other than it is NP-hard to approximate [1]. One special case which receives many attention is $k$-complex contagion where a node becomes active if at least $k$ of its neighbours have been activated [22, 23, 24]. Gao et al. [25] make one step further of $k$-complex contagion model by considering the threshold comes from a probability distribution.

Optimization of non-submodular function is another interesting direction. Du et al. [26] introduce two techniques — restricted submodularity and shifted submodularity — to analyze greedy approximation of non-submodular functions. Recently, Horel et al.[27] study the problem of maximizing a set function that is very close to submodular. They assume that function values can be obtained from an oracle and focused on its query complexity. In our study, the local threshold functions are close to submodular and our target is to study its effect on the global influence spread function, which is the result of complex cascading behavior derived from the local threshold functions.

## 2 Preliminaries

For a set function $f : 2^V \to \mathbb{R}$, we say that it is *monotone* if $f(S) \leq f(T)$ for all $S \subseteq T \subseteq V$; we say that it is *submodular* if $f(S \cup \{v\}) - f(S) \geq f(T \cup \{v\}) - f(T)$, for all $S \subseteq T \subseteq V$ and $v \in V \setminus T$. For a directed graph $G = (V, E)$, we use $N^{in}(v)$ to denote the in-neighbors of $v$ in $G$. We now formally define the general threshold model used in the paper.

**Definition 1** (General Threshold Model [1]). *In the general threshold model, for a social graph $G = (V, E)$, every node $v \in V$ has a threshold function $f_v : 2^{N^{in}(v)} \to [0, 1]$. The function $f_v(\cdot)$ should be monotone and $f_v(\emptyset) = 0$. Initially at time 0, each node $v \in V$ is in the inactive state and chooses $\theta_v$ uniformly at random from the interval $[0, 1]$. A seed set $S_0$ is also selected, and their states are set to be active. Afterwards, the influence propagates in discrete time steps. At time step $t \geq 1$, node $v$ becomes active if $f_v(S_{t-1} \cap N^{in}(v)) \geq \theta_v$, where $S_{t-1}$ is the set of active nodes by time step $t - 1$. The process ends when no new node becomes active in a step.*

General threshold model is one of the most important models in the influence maximization problem. Usually we focus on two properties of threshold function – submodularity and supermodularity. Submodularity can be understood as diminishing marginal returns when adding more nodes to the seed set. In contrast, supermodularity means increasing marginal returns. Given a seed set $S$, let $\sigma(S)$ denote the expected number of activated nodes after the process of influence propagation terminates, and we call $\sigma(S)$ the *influence spread* of $S$.

Submodularity is the key property that guarantees the performance of greedy algorithms [9]. In this paper, we would like to study the influence maximization with nearly submodular threshold function — $\varepsilon$-almost submodular function, or in short $\varepsilon$-AS.

**Definition 2** ($\varepsilon$-Almost Submodular ($\varepsilon$-AS)). *A set function $f : 2^V \to \mathbb{R}$ is $\varepsilon$-almost submodular if there exists a submodular function $f^{sub}$ defined on $2^V$ and for any subset $S \subseteq V$, $f^{sub}(S) \geq f(S) \geq (1 - \varepsilon)f^{sub}(S)$. Here $\varepsilon$ is a small positive number.*

The definition of $\varepsilon$-almost submodular here is equivalent to "Approximate submodularity" defined in [27]. For an $\varepsilon$-almost submodular threshold function $f_v$, define its upper and lower submodular bound as $\overline{f}_v$ and $\underline{f}_v$. Hence by definition, we have $\underline{f}_v = (1 - \varepsilon)\overline{f}_v$. Given the definition of $\varepsilon$-almost submodular function, we then model the *almost submodular* graph. In this paper, we consider the influence maximization problem based on this kind of graphs.

**Definition 3** (($\gamma, \varepsilon$)-Almost Submodular Graph). *Given fixed parameters $\gamma, \varepsilon \in [0, 1]$, we say that a graph with $n$ ($n = |V|$) nodes is a $(\gamma, \varepsilon)$-Almost Submodular Graph (under the general threshold model), if there are at most $n^\gamma$ nodes in the graph with $\varepsilon$-almost submodular threshold functions while other nodes have submodular threshold functions.*

**Definition 4** ($\varepsilon$-ASIM). *Given a graph containing $\varepsilon$-almost submodular nodes and an input $k$, Influence Maximization problem on graph with $\varepsilon$-Almost Submodular nodes ($\varepsilon$-ASIM) is the problem to find $k$ seed nodes such that the influence spread invoked by the $k$ nodes is maximized.*

## 3 Inapproximability of $\varepsilon$-ASIM

In this section we show that it is in general hard to approximate the influence maximization problem even if there are only sublinear number of nodes with $\varepsilon$-almost submodular threshold functions. The main reason is that even a small number of nodes with $\varepsilon$-almost submodular threshold functions $f_v(\cdot)$ would cause the global influence spread function far from submodularity, making the maximization problem very difficult. The theorem below shows our hardness result.

**Theorem 1.** *For any small $\varepsilon > 0$ and any $\gamma \in (0, 1)$, there is no $1/n^{\frac{\gamma}{c}}$-approximation influence maximization algorithm for all $(\gamma, \varepsilon)$-almost submodular graphs where $c = 3 + 3/\log \frac{2}{2-\varepsilon}$, unless P=NP.*

We first construct a probabilistic-AND gate gadget by amplifying the non-submodularity through a binary tree. Then we prove the lower bound of approximation ratio by the reduction from set cover problem. Due to page limits, we only sketch the main technique. The proof of Theorem 1 is given in Appendix **??**.

Here we construct a basic gadget with input $s_1, s_2$ and output $t$ (see Figure 1a). We assume that node $t$ has two in-neighbours $s_1, s_2$ and the threshold function $g(\cdot)$ of $t$ is $\varepsilon$-almost submodular: $g(S) = |S|/2$, when $|S| = 0$ or $2$; $\frac{1-\varepsilon}{2}$, when $|S| = 1$.

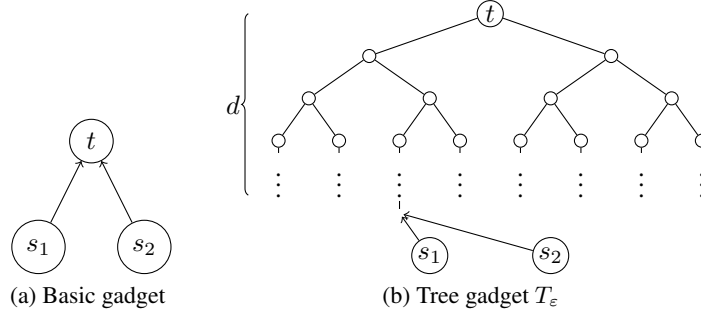

(a) Basic gadget             (b) Tree gadget $T_\varepsilon$

Figure 1: Diagrams of gadegts

Let $P_a(v)$ be the activation probability of node $v$ in this case. This simple gadget is obviously far away from the AND gate, and our next step is to construct a more complex gadget with input node $s_1, s_2$. We hope that the output node $t$ is active only when both $s_1, s_2$ are active, and if only one of $s_1$ and $s_2$ is active, the probability that node $t$ becomes active is close to $0$. We call it a probabilistic-AND gate.

The main idea is to amplify the gap between submodularity and non-submodularity by binary tree (figure 1b). In this gadget $T_\varepsilon$ with a complete binary tree, node $t$ is the root of a full binary tree and each node holds a directed edge to its parent. For each leaf node $v$ in the tree, both $s_1, s_2$ hold the directed edges towards it. The threshold function for each node in the tree is $g(\cdot)$ defined above while $\varepsilon$ is the index of gadget $T_\varepsilon$. The depth of the tree is parameter $d$ which will be determined later. We use $v_i$ to denote a node of depth $i$ ($t$ is in depth 1). It is obviously that $P_a(t) = 1$ if both $s_1, s_2$ are activated, and $P_a(t) = 0$ if neither $s_1$ or $s_2$ is activated. Thus, we would like to prove, in case when only one of $s_1, s_2$ is activated, the activation probability becomes smaller for inner nodes in the tree.

**Lemma 2.** *For gadget $T_\varepsilon$ with depth $d$, the probability of activating output node $t$ is less than $(\frac{2-\varepsilon}{2})^d$ when only one of $s_1, s_2$ is activated.*

*Proof.* In this case, for leaf node $v_d$, we have $P_a(v_d) = \frac{1-\varepsilon}{2}$. Apparently, the probability of becoming activated for nodes with depth $d$ are independent with each other. Given a basic gadget, if each of the two children nodes is activated with an independent probability $p$, then the parent node will be activated with probability

$$p^2 \times g(2) + 2p(1-p) \times g(1) + (1-p)^2 \times g(0) = p^2 + 2p(1-p)\frac{1-\varepsilon}{2} = p(1 - \varepsilon(1-p)).$$

So we have $P_a(v_i) \leq P_a(v_{i+1})(1 - \varepsilon(1 - P_a(v_{i+1})))$. Since $P_a(v_d) = \frac{1-\varepsilon}{2} < 1/2$, and $P_a(v_i) \leq P_a(v_{i+1})$ from above, we have $p_a(v_i) < 1/2$ for all $i$, and thus we can rewrite the recurrence as $P_a(v_i) \leq P_a(v_{i+1})(1 - \varepsilon/2)$. Hence for the gadget with depth $d$, the probability that node $t$ becomes activated is $P_a(t) = P_a(v_1) \leq \frac{1-\varepsilon}{2}(\frac{2-\varepsilon}{2})^{d-1} < (\frac{2-\varepsilon}{2})^d$. $\qquad\square$

Lemma 2 shows that gadget $T_\varepsilon$ is indeed a *probabilistic-AND gate* with two input nodes, and the probability that $t$ is activated when only one of $s_1$ and $s_2$ is activated approaches $0$ exponentially fast with the depth $d$. We say a gadget $T_\varepsilon$ works well if output node $t$ stay inactive when only one of the input nodes is activated.

By the similar method we construct multi-input-AND gates based on 2-input-AND gates. Finally, we show that if the influence maximization problem can be approximated beyond the ratio shown above, we can solve the set cover problem in polynomial time. The main idea is as follows. For any set cover instance, we will put all elements to be the input of our multi-input-probabilistic-AND gate, and connect the output with a large number of additional nodes. Thus, if $k$ sets can cover all elements, all of those addition nodes will be activated, on contrast, if at least one of the elements cannot be covered, almost all of the additional nodes will remain inactive.

# 4 Approximation Algorithms

In the previous section, we show that influence maximization is hard to approximate when the number of $\varepsilon$-almost submodular nodes is sublinear but still a non-constant number. In this section, we discuss the situation where only small number of nodes hold $\varepsilon$-almost submodular threshold functions. We firstly provide a greedy algorithm for small number of non-submodular nodes which may not be $\varepsilon$-almost submodular, then, we restrict to the case of $\varepsilon$-almost submodular nodes.

## 4.1 Approximation Algorithm with Small Number of Non-submodular Nodes

In the case of $\ell$ ($\ell < k$) non-submodular nodes, we provide an approximate algorithm as follows. We first add these non-submodular nodes into the seed set, and then generate the rest of the seed set by the classical greedy algorithm. The proof of Theorem 3 can be found in Appendix **??**.

**Theorem 3.** *Given a graph of $n$ nodes where all nodes have submodular threshold functions except $\ell < k$ nodes, for influence maximization of $k$ seeds with greedy scheme we can obtain a $(1 - e^{-\frac{k-\ell}{k}})$-approximation ratio.*

## 4.2 Approximation Algorithm of $\varepsilon$-ASIM

In this section, we consider the case when all non-submodular nodes have $\varepsilon$-almost submodular threshold functions, and provide an approximation algorithm that allows more than $k$ $\varepsilon$-almost submodular nodes, with the approximation ratio close to $1 - 1/e$ when $\varepsilon$ is small. The main idea is based on the mapping between probability spaces.

Given a graph containing nodes with $\varepsilon$-almost submodular threshold functions, we simply set each node's threshold function to its submodular lower bound and then run classical greedy algorithm $\mathcal{A}$ on this graph (Algorithm 1). Algorithm 1 takes the lower bounds of $\varepsilon$-almost submodular threshold functions as input parameters. The following theorem analyzes the performance of Algorithm 1.

---

**Algorithm 1 Galg-L** algorithm for Influence Maximization

---

**Input:** $G = (V, E)$, $\mathcal{A}$, $\{f_v\}, \{\underline{f}_v\}$, $k$
**Output:** Seed set $S$
 1: set $S = \emptyset$
 2: replace each nodes $v$'s threshold function $f_v$ with $\underline{f}_v$
 3: run algorithm $\mathcal{A}$ on $G$ with $\{\underline{f}_v\}$ and obtain $S$
 4: **return** $S$

---

**Theorem 4.** *Given a graph $G = (V, E)$, under the general threshold model, assuming that $\ell$ nodes have $\varepsilon$-almost submodular threshold functions and the other nodes have submodular threshold functions. Then the greedy algorithm **Galg-L** has approximation ratio of $(1 - \frac{1}{e})(1 - \varepsilon)^\ell$.*

*Proof.* Let $V_e$ be the set of nodes with $\varepsilon$-almost submodular threshold functions. Without loss of generality, we assume $V_e = \{v_1, v_2, \ldots, v_\ell\}$. Now consider two general threshold models $\overline{M}, \underline{M}$ with different threshold functions. Both models hold threshold functions $\{f_v\}$ for $v \in V - V_e$. For node $v$ in $V_e$, $\overline{M}, \underline{M}$ hold $\{\overline{f}_v\}$ and $\{\underline{f}_v\}$ respectively.

In any threshold model, after we sample each node's threshold $\theta_v$, the diffusion process becomes deterministic. A graph with threshold functions $\{f_v\}$ and sampled thresholds $\{\theta_v\}$ is called a *possible world* of the threshold model, which is similar to the live-edge graph in the independent cascade model. An instance of threshold model's possible world can be written as $\{\theta_{v_1}, \theta_{v_2}, \ldots, \theta_{v_n}; f_{v_1}, f_{v_2}, \ldots, f_{v_n}\}$. Here we build a one-to-one mapping from all $\underline{M}$'s possible worlds with $\theta_v \leq 1 - \varepsilon$ ($v \in V_e$) to all $\overline{M}$'s possible worlds:

$$\{\theta_{v_1}, \ldots, \theta_{v_n}; f_{v_1}, \ldots, f_{v_n}\} \leftrightarrow \{\frac{\theta_{v_1}}{1-\varepsilon}, \ldots, \frac{\theta_{v_\ell}}{1-\varepsilon}, \theta_{v_{\ell+1}} \ldots, \theta_{v_n}; \frac{f_{v_1}}{1-\varepsilon}, \ldots, \frac{f_{v_\ell}}{1-\varepsilon}, f_{v_{\ell+1}}, \ldots, f_{v_n}\}.$$

The above corresponding relation shows this one-to-one mapping between $\underline{M}$ and $\overline{M}$. For any instance of $\underline{M}$'s possible world with $\theta_v \leq 1 - \varepsilon$ ($v \in V_e$), we amplify the threshold of node $v$ in $V_e$ to $\frac{\theta_v}{1-\varepsilon}$. At the same time, we amplify the corresponding threshold function by a factor of $\frac{1}{1-\varepsilon}$.

Obviously, this amplification process will not effect the influence process under this possible world, because for each $v \in V_e$, both its threshold value and the its threshold function are amplified by the same factor $1/(1-\varepsilon)$. Furthermore, the amplified possible world is an instance of $\overline{M}$.

Expected influence can be computed by $\sigma(S) = \int_{\vec{\theta} \in [0,1]^n} \mathcal{D}(\vec{\theta}; \vec{f}, S) \mathrm{d}_{\vec{\theta}}$, where $\mathcal{D}(\vec{\theta}; \vec{f}, S)$ is the deterministic influence size of seed set $S$ under possible world $\{\vec{\theta}; \vec{f}\}$. We refer $\overline{M}, \underline{M}$'s expected influence size functions as $\overline{\sigma}, \underline{\sigma}$. We define $\vec{\theta} \in [0,1]^n$ as the vector of $n$ nodes threshold, and $\vec{\theta}_e \in [0,1]^\ell$, $\vec{\theta}' \in [0,1]^{n-\ell}$ are the threshold vectors of $V_e$ and $V - V_e$. Besides, the threshold functions of $V_e$ and $V - V_e$ will be represented as $\vec{f}_e, \vec{f}'$. A possible world is symbolized as $\{\vec{\theta}_e, \vec{\theta}'; \vec{f}_e, \vec{f}'\}$. For any seed set $S$, we have

$$
\begin{aligned}
\underline{\sigma}(S) &= \int_{\vec{\theta} \in [0,1]^n} \mathcal{D}(\vec{\theta}; \vec{f}, S) \mathrm{d}_{\vec{\theta}} \\
&\geq \int_{\vec{\theta}_e \in [0,1-\varepsilon]^\ell} \int_{\vec{\theta}' \in [0,1]^{n-\ell}} \mathcal{D}((\vec{\theta}_e, \vec{\theta}'); \vec{f}, S) \mathrm{d}_{\vec{\theta}_e} \mathrm{d}_{\vec{\theta}'} \\
&= (1-\varepsilon)^\ell \int_{\frac{\vec{\theta}_e}{1-\varepsilon} \in [0,1]^\ell} \int_{\vec{\theta}' \in [0,1]^{n-\ell}} \mathcal{D}((\vec{\theta}_e, \vec{\theta}'); \vec{f}, S) \mathrm{d}_{\frac{\vec{\theta}_e}{1-\varepsilon}} \mathrm{d}_{\vec{\theta}'} \\
&= (1-\varepsilon)^\ell \int_{\frac{\vec{\theta}_e}{1-\varepsilon} \in [0,1]^\ell} \int_{\vec{\theta}' \in [0,1]^{n-\ell}} \mathcal{D}((\frac{\vec{\theta}_e}{1-\varepsilon}, \vec{\theta}'); (\frac{\vec{f}_e}{1-\varepsilon}, \vec{f}'), S) \mathrm{d}_{\frac{\vec{\theta}_e}{1-\varepsilon}} \mathrm{d}_{\vec{\theta}'} \\
&= (1-\varepsilon)^\ell \int_{\vec{\theta} \in [0,1]^n} \mathcal{D}(\vec{\theta}; (\frac{\vec{f}_e}{1-\varepsilon}, \vec{f}'), S) \mathrm{d}_{\vec{\theta}} \\
&= (1-\varepsilon)^\ell \overline{\sigma}(S).
\end{aligned}
$$

The third equality utilizes our one-to-one mapping, in particular $\mathcal{D}((\vec{\theta}_e, \vec{\theta}'); \vec{f}, S) = \mathcal{D}((\frac{\vec{\theta}_e}{1-\varepsilon}, \vec{\theta}'); (\frac{\vec{f}_e}{1-\varepsilon}, \vec{f}'), S)$ for $\frac{\vec{\theta}_e}{1-\varepsilon} \in [0,1]^\ell$, because they follow the same deterministic propagation process. Hence given a seed set $S$, the respective influence sizes in model $\overline{M}, \underline{M}$ satisfy the relation $\underline{\sigma}(S) \geq (1-\varepsilon)^\ell \overline{\sigma}(S)$.

Let $\sigma$ be the expected influence size function of the original model, and assume that the optimal solution for $\overline{\sigma}, \sigma, \underline{\sigma}$ are $\overline{S}^*, S^*, \underline{S}^*$ respectively. Apparently, $\overline{\sigma}(\overline{S}^*) \geq \sigma(S^*)$ since for every node $v$, $\overline{f}_v \geq f_v$. According to the previous analysis, we have $\underline{\sigma}(\underline{S}^*) \geq \underline{\sigma}(\overline{S}^*) \geq (1-\varepsilon)^\ell \overline{\sigma}(\overline{S}^*)$. Hence for output $S^{\mathcal{A}}$ of the greedy algorithm for optimizing $\underline{\sigma}$, we have approximation ratio

$$
\sigma(S^{\mathcal{A}}) \geq \underline{\sigma}(S^{\mathcal{A}}) \geq (1 - \frac{1}{e})\underline{\sigma}(\underline{S}^*) \geq (1 - \frac{1}{e})(1-\varepsilon)^\ell \overline{\sigma}(\overline{S}^*) \geq (1 - \frac{1}{e})(1-\varepsilon)^\ell \sigma(S^*).
$$

The theorem holds. $\qquad\square$

If we replace threshold functions by their upper bound and run the greedy algorithm, we obtain **Galg-U**. With similar analysis, **Galg-U** also holds approximation ratio of $(1 - \frac{1}{e})(1-\varepsilon)^\ell$ on graphs with $\ell$ $\varepsilon$-almost submodular nodes. The novel technique used to prove approximation ratio is similar to the *sandwich approximation* in [28]. But their approximation ratio relies on instance-dependent influence sizes, while we utilize mapping of probabilistic space to provide instance-independent approximation ratio.

## 5 Experiments

In addition to the theoretical analysis, we are curious about the performance of greedy algorithms **Galg-U, Galg-L** on real networks with non-submodular nodes. Our experiments run on a machine with two 2.4GHz Intel(R) Xeon(R) E5-2620 CPUs, 4 processors (24 cores), 128GB memory and 64bit Ubuntu 14.04.1. All algorithms tested in this paper are written in C++ and compiled with g++ 4.8.4. Some algorithms are implemented with multi-thread to decrease the running time.

### 5.1 Experiment setup

**Datasets.** We conduct experiments on three real networks. The first network is NetHEPT, an academic collaboration network extracted from "High Energy Physics - Theory" section of arXiv (http://www.arXiv.org) used by many works [7, 14, 15, 19, 20]. NetHEPT is an undirected network with 15233 nodes and 31376 edges, each node represents an author and each edge represents that two authors collaborated on a paper. The second one is Flixster, an American movie rating social site. Each node represents a user, and directed edge $(u, v)$ means $v$ rated the same movie shortly after $u$

did. We select topic 3 with 29357 nodes and 174939 directed edges here. The last one is the DBLP dataset, which is a larger collaboration network mined from the computer science bibliography site DBLP with 654628 nodes and 1990259 undirected edges [14]. We process its edges in the same way as the NetHEPT dataset.

**Propagation Models.** We adapt general threshold model in this paper. Our Galg-U,Galg-L are designed on submodular upper and lower bounds, respectively. Since directly applying greedy scheme on graphs with submodular threshold function is time-consuming, we assign the submodular threshold function and submodular upper bound of $\varepsilon$-AS function as linear function here: $f_v(S) = |S|/d(v)$, where $d(v)$ is the in-degree of $v$. This makes the corresponding model an instance of the linear-threshold model, and thus the greedy algorithm can be accelerated with Reverse Reachable Set (RRset) technique [17].

We construct two different $\varepsilon$-almost submodular threshold functions in this paper: (1) a power function $\frac{|S|}{d(v)}^{\beta}$ with $\beta$ satisfying $\frac{1}{d(v)}^{\beta} = \frac{1}{d(v)}(1 - \varepsilon)$; (2) $f_v(S) = \frac{|S|}{d(v)}(1 - \varepsilon)$ for $|S| \leq 2$ and $|S|/d(v)$ otherwise. The former $\varepsilon$-almost submodular function is a supermodular function. The supermodular phenomenon has been observed in Flickr [11]. The second $\varepsilon$-almost submodular function is just dropping down the original threshold function for the first several nodes, which is consistent with the phenomenon observed in LiveJournal [10]. We call them $\varepsilon$-AS-1 and $\varepsilon$-AS-2 functions respectively.

**Algorithms.** We test our approximation Algorithm 1 and other baseline algorithms on the graphs with $\varepsilon$-almost submodular nodes.

- TIM-U, TIM-L: Tang et al. [17] proposed a greedy algorithm TIM$^+$ accelerated with Reverse Reachable Set (RRset). The running time of TIM$^+$ is $O(k(m + n) \log n)$ on graphs with $n$ nodes and $m$ edges. RRset can be sampled in live-edge graph of IC model, and with some extension we can sample RRset under Triggering model [1]. LT model also belongs to Triggering model, but General Threshold model with non-submodular threshold functions does not fall into the category of Triggering model. Thus TIM$^+$ can not be directly applied on original graphs with non-submodular nodes. In our experiments, we choose $\varepsilon$-AS-1 and $\varepsilon$-AS-2 thresholds to ensure that TIM$^+$ can run with their upper or lower bound. We then run Algorithm 1 with TIM$^+$ as input. Algorithm Galg-L based on TIM$^+$ can be written in short as TIM-L. By using the upper bound we obtain TIM-U.

- Greedy: We can still apply the naive greedy scheme on graph with $\varepsilon$-almost submodular nodes and generate results without theoretical guarantee. The naive greedy algorithm is time consuming, with running time is $O(k(m + n)n)$.

- High-degree: High-degree outputs seed set according to the decreasing order of the out-degree.

- PageRank: PageRank is widely used to discover nodes with high influence. The insight of PageRank is that important nodes point to important nodes. In this paper, The transition probability on edge $e = (u, v)$ is $1/d(u)$. We set restart probability as $0.15$ and use the power method to compute the PageRank values. Finally PageRank outputs nodes with top PageRank values.

- Random: Random simply selects seeds randomly from node set.

**Experiment methods.** The datasets provide the structure of network, and we first assume each node holds linear threshold function as described above. In each experiment, we randomly sample some nodes with in-degree greater than 2, and assign those nodes with our $\varepsilon$-almost submodular functions, $\varepsilon$-AS-1 or $\varepsilon$-AS-2. Since the naive greedy algorithm is quite time-consuming, we just run it on NetHEPT.

### 5.2 Experiment results

**Results on NetHEPT.** Our first set of experiments focuses on the NetHEPT dataset with the aim of comparing TIM-U, TIM-L and Greedy. TIM-U, TIM-L have theoretical guarantee, but the approximation ratio is low when the graph contains a considerable number of $\varepsilon$-AS nodes. Figure 2 shows the influence size of each method, varying from 1 to 100 seeds. Figure 2a and 2b are results conducted on constructed graph with $\varepsilon$-AS-1 nodes. Observe that TIM-U, TIM-L slightly outperform Greedy in all cases. Compared with results of 3000 $\varepsilon$-AS nodes, influence of output seeds drops obviously in graph with 10000 $\varepsilon$-AS nodes. But the ratio that TIM-U, TIM-L exceed PageRank

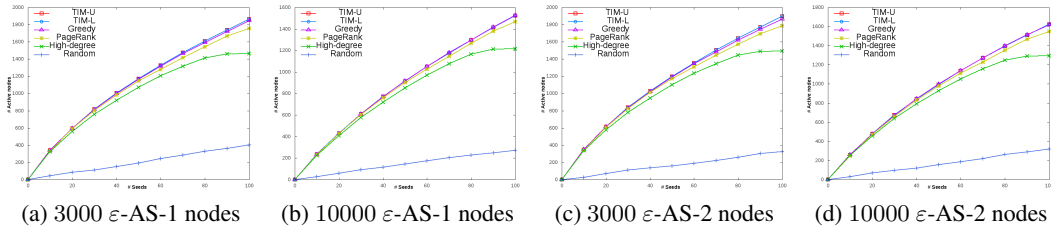

| (a) 3000 $\varepsilon$-AS-1 nodes | (b) 10000 $\varepsilon$-AS-1 nodes | (c) 3000 $\varepsilon$-AS-2 nodes | (d) 10000 $\varepsilon$-AS-2 nodes |

Figure 2: Results of IM on NetHEPT with $\varepsilon = 0.2$

increases with rising fraction of $\varepsilon$-AS nodes. In particular, $\varepsilon$-AS-1 is indeed supermodular, TIM-U, TIM-L beats Greedy even when many nodes have supermodular threshold functions.

We remark that TIM-U, TIM-L and Greedy outperform other baseline algorithms significantly. When $k = 100$, TIM-U is $6.1\%$ better compared with PageRank and $27.2\%$ better compared with High-degree. When conducted with $\varepsilon$-AS-2 function, Figure 2c and 2d report that TIM-U, TIM-L and Greedy still perform extremely well. Influence size conducted on graphs with $\varepsilon$-AS-2 function is better than those with $\varepsilon$-AS-1 function. This is what we expect: supermodular function is harder to handle among the class of $\varepsilon$-almost submodular functions.

Another thing to notice is that TIM-U, TIM-L can output seeds on NetHEPT within seconds, while it takes weeks to run the naive greedy algorithm. With RRsets technique, TIM$^+$ dramatically reduces the running time. The $\varepsilon$-almost submodular functions selected here ensure that TIM$^+$ can be invoked. Since TIM-U, TIM-L match the performance of Greedy while TIM-U, TIM-L are scalable, we do not run Greedy in the following larger datasets.

**Results on Flixster.** Figure 3 shows the results of experiments conducted on Flixster with

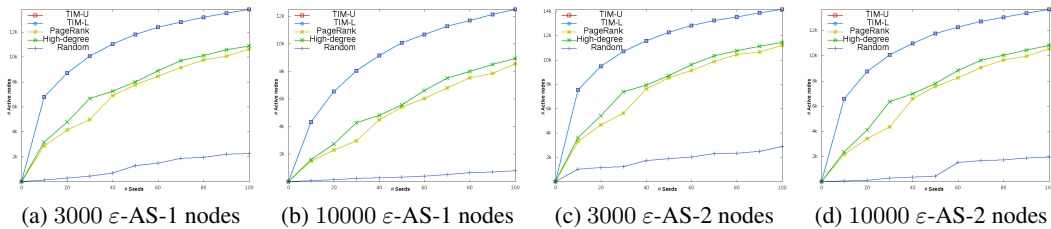

| (a) 3000 $\varepsilon$-AS-1 nodes | (b) 10000 $\varepsilon$-AS-1 nodes | (c) 3000 $\varepsilon$-AS-2 nodes | (d) 10000 $\varepsilon$-AS-2 nodes |

Figure 3: Results of IM on Flixster with $\varepsilon = 0.2$

$\varepsilon = 0.2$. We further evaluate algorithms by Flixster with $\varepsilon = 0.4$ (see Figure 4). Observe that TIM-U, TIM-L outperform other heuristic algorithms in all cases. Compared with PageRank, $30\%, 46.3\%, 26\%, 29.7\%$ improvement are observed in the four experiments in Figure 3. TIM-U performs closely to TIM-L consistently. The improvement is larger than that in NetHEPT. The extra improvement might due to more complex network structure. The average degree is $5.95$ in Flixster, compared to $2.05$ in NetHEPT. In dense network, nodes may be activated by multiple influence chains, which makes influence propagates further from seeds. Baseline algorithms only pay attention to the structure of the network, hence they are defeated by TIM-U, TIM-L that focus on influence spread. The more $\varepsilon$-AS nodes in network, the more improvement is obtained.

When we set $\varepsilon$ as $0.4$, Figure 4 shows that TIM-U is $37.6\%, 74.2\%, 28\%, 35.6\%$ better than PageRank respectively. Notice that the gap between the performances of TIM-U and PageRank increases as $\varepsilon$ increases. In Flixster dataset, we observe that TIM-U,TIM-L hold greater advantage in case of larger number of $\varepsilon$-AS nodes and larger $\varepsilon$.

**Results on DBLP.** For DBLP dataset, the results are shown in Figure 5. TIM-U and TIM-L are still the best algorithms according to performance. But PageRank and High-degree also performs well, just about $2.6\%$ behind TIM-U and TIM-L. DBLP network has many nodes with large degree, which correspond to those active scientists. Once such active authors are activated, the influence will increase significantly. This may partly explain why TIM-U,TIM-L perform similarly to PageRank.

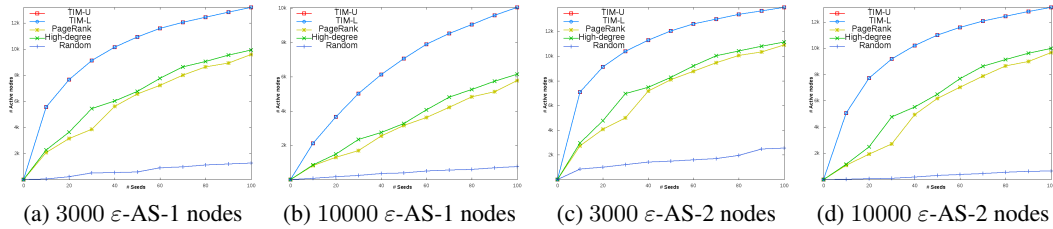

(a) 3000 $\varepsilon$-AS-1 nodes    (b) 10000 $\varepsilon$-AS-1 nodes    (c) 3000 $\varepsilon$-AS-2 nodes    (d) 10000 $\varepsilon$-AS-2 nodes

Figure 4: Results of IM on Flixster with $\varepsilon = 0.4$

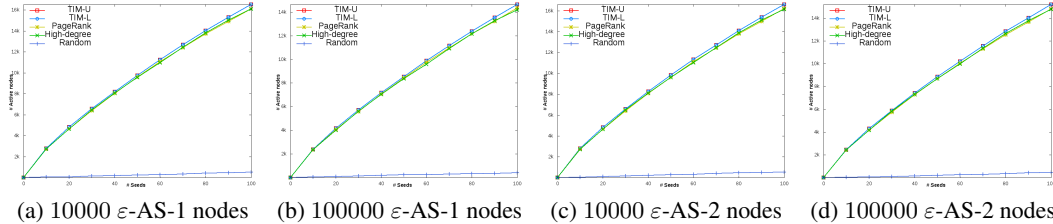

(a) 10000 $\varepsilon$-AS-1 nodes    (b) 100000 $\varepsilon$-AS-1 nodes    (c) 10000 $\varepsilon$-AS-2 nodes    (d) 100000 $\varepsilon$-AS-2 nodes

Figure 5: Results of IM on DBLP with $\varepsilon = 0.2$

## 6 Conclusion and Future Work

In this paper, we study the influence maximization problem on propagation models with non-submodular threshold functions, which are different from most of existing studies where the threshold functions and the influence spread function are both submodular. We investigate the problem by studying a special case — the $\varepsilon$-almost submodular threshold function. We first show that influence maximization problem is still hard to approximate even when the number of $\varepsilon$-almost submodular nodes is sub-linear. Next we provide a greedy algorithm based on submodular lower bounds of threshold function to handle the graph with small number of $\varepsilon$-almost submodular nodes and show its theoretical guarantee. We further conduct experiments on real networks and compare our algorithms with other baselines to evaluate our algorithms in practice. Experimental results show that our algorithms not only have good theoretical guarantees on graph with small number of $\varepsilon$-almost submodular nodes, they also perform well on graph with a fairly large fraction of $\varepsilon$-almost submodular nodes.

Our study mainly focuses on handling $\varepsilon$-almost submodular threshold functions. One future direction is to investigate models with arbitrary non-submodular threshold functions. Another issue is that the greedy algorithms we propose are slow when the submodular upper bound or lower bound of threshold function do not correspond to the Triggering model. It remains open whether we could utilize RRset or other techniques to accelerate our algorithms under this circumstance. How to accelerate the naive greedy process with arbitrary submodular threshold functions is another interesting direction.

### Acknowledgments

This work was supported in part by the National Natural Science Foundation of China Grant 61433014, 61502449, 61602440, the 973 Program of China Grants No. 2016YFB1000201.

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
