[Supplementary Material]

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

## Supplementary Material

## A    Missing Proof of Theorem 1

*Proof.* We have described the construction of gadget $T_\varepsilon$, we can further construct $n$-input trees $T_\varepsilon^n$ with gadget $T_\varepsilon$ — a multi input AND gate. Given $n$ input nodes $s_1, s_2, \ldots, s_n$, we can use $s_0$ and $s_1$ as input nodes of $T_\varepsilon$ and set output node as $s_{12}$. And then combine $s_{12}$ and $s_3$ with gadget $T_\varepsilon$ to obtain output $s_{123}$. Finally we obtain ultimate output node $s_{12\ldots n}$. With high probability $s_{12\ldots n}$ will not be activated if not all input nodes $s_1, s_2, \ldots, s_n$ are activated. We calculate the probability that $T_\varepsilon^n$ works well. Since each $T_\varepsilon$ is a binary tree with depth $d$, according to Lemma 2 each $T_\varepsilon$ breaks the AND gate rule with probability at most $(\frac{2-\varepsilon}{2})^d$. With Union Bound, we can say that $T_\varepsilon^n$ is an AND gate with probability at least $1 - n(\frac{2-\varepsilon}{2})^d$. Besides, a $T_\varepsilon^n$ have $n \times (2^d - 1) = n2^d - n$ nodes.

We consider a variant of set cover problems and show that if influence maximization problem can be approximated beyond the ratio shown above, then we can solve the set cover problem. Let $e_1, e_2, \ldots, e_n$ denote the nodes corresponding to the $n$ elements. Let $s_1, s_2, \ldots, s_m$ denote the nodes corresponding to the $m$ sets. We can assume that $m < n$. When $m \geq n$ we could add $m$ dummy nodes that contained by all $m$ sets, the solution of the newly created graph is the same with original graph. Note that in this reduction, $n$ is the number of elements in the Set Cover instance. We use $N$ to denote the size of the graph constructed from the Set Cover instance.

Figure 6 shows the graph structure for the reduction. Each set node $s_i$ holds a directed edge towards the element nodes it covers and each element node $e_j$ has threshold function with $f_{e_j}(1) = 1$, which means that $e_j$ becomes activated when at least one of the nodes corresponding to sets covering it is activated. We add $n^\alpha$ copies of $T_\varepsilon^n$ trees, each of which has $e_1, e_2, \ldots, e_n$ as the input nodes, and $x_i$ as the output node. Moreover, for a large positive constant $\beta$, we add $n^\beta$ children $c_1, c_2, \ldots, c_{n^\beta}$ for each node $x_k$. Each child node $c_l$ also has threshold function with $f_{c_l}(1) = 1$. That is, these $c_l$ nodes have submodular threshold functions. For the constructed graph, only nodes in $T_\varepsilon^n$ (nodes painted red) holds $\varepsilon$-almost submodular threshold function. We can adjust the ratio of $\varepsilon$-almost submodular nodes with the assignment of $\alpha, \beta$. Intuitively, to make the approximation ratio to be proved in the hardness result at a low value of $1/N^a$ for some $a > 0$, we want to increase $\alpha$ to create a larger gap between the existence and non-existence of a set cover. In particular, the fraction of nodes that could be activated by $k$ seeds in the case of no set cover should be less than the approximation ratio $1/N^a$. However, increasing $\alpha$ would generate more $\varepsilon$-almost submodular nodes. To keep the fraction of $\varepsilon$-almost submodular nodes small, we need to increase $\beta$. But increasing $\beta$ would increase the total size of the graph $N$, decreasing the approximation ratio $1/N^a$. Therefore, we need to find the right balance between $\alpha$ and $\beta$ to achieve what we claim in Theorem 1. The following is the detailed analysis of setting parameters $\alpha$ and $\beta$ and the depth of trees $T_\varepsilon^n$.

Figure 6: Reduction structure from a Set Cover instance to an $\varepsilon$-ASIM instance

Here we calculate the depth of $T_\varepsilon$ in order to make sure that the $n^\alpha$ $T_\varepsilon^n$ are all AND gate with high probability. Again we apply Union Bound to calculate the probability upper bound of at least one $T_\varepsilon^n$ fails — $n^{1+\alpha}(\frac{2-\varepsilon}{2})^d$. We set $d = (1+\alpha+\lambda)\log n/\log\frac{2}{2-\varepsilon}$, so all the $T_\varepsilon^n$ observe the rule of AND gate with probability at least $1 - n^{-\lambda}$.

If we can find $k$ sets that cover all the $n$ elements, then we just select the $k$ nodes corresponding to the $k$ sets as seed nodes. Then the seed nodes will activate nodes $e_1, e_2, \ldots, e_n$ and then all nodes in gadgets $T_\varepsilon^n$ will become activated. Finally, all the child nodes of $x_1, x_2, \ldots, x_{n^\alpha}$ become activated, and totally $k + n + n^{\alpha+1}(2^d - 1) + n^{\alpha\beta} \geq n^{\alpha\beta}$ nodes become activated in total, while the graph consists of $N = m + n + n^{\alpha+1}(2^d - 1) + n^{\alpha\beta}$ nodes, nearly all nodes are activated. On the other hand, if there is no solution of size $k$ for this set cover problem, we can not activate all element nodes $e_j$. During the influence spread of a given target nodes, with probability al least $1 - n^{-\lambda}$ none of $x_1, x_2, \ldots, x_{n^\alpha}$ will become activated. Under this circumstance, we can select nodes labeled with $x$ as seeds or just try to activate more nodes $e_j$. Totally we can just activate at most $k + n + kn^\beta$ nodes if we choose $x_k$ as seeds. Otherwise we can assume that at most $n - 1$ elements are covered and all gadget nodes are activated. In this case $k + n - 1 + n^{\alpha+1}(2^d - 1)$ nodes will become active if all probabilistic AND gate work well. On the other hand if these $T_\varepsilon^n$ fails with probability at most $n^{-\lambda}$, we just assume that all the nodes will become activated eventually. We can set $n^\beta = n^{\alpha-1} \cdot 2^d \cdot n^\delta$ for $\delta > 0$ to ensure that the fraction of gadget nodes and dummy nodes is slightly less than $n^{\frac{1}{\alpha-1}}$. When $n$ is large enough the influence size we obtain that influence size is less then $kn^\beta \leq n^{\beta+1}$ or $n^{\alpha+1}2^d$ w.h.p. when Set Cover is not solved. For any influence maximization algorithm, there exists a graph of $N$ nodes where at most $n^{\alpha+1}(2^d - 1)$ nodes have $\varepsilon$-almost submodular threshold function, for any result obtained, with probability at least $1 - n^{-\lambda}$, the influence size is less than $1/n^{\alpha-1}$ of $\sigma(S^*)$, unless we can solve Set Cover problems within polynomial time. Here $N \geq n^{\alpha+\beta}$, $d = (1+\alpha+\lambda)\log n/\log\frac{2}{2-\varepsilon}$, $\beta = (\alpha+1) + (1+\alpha+\lambda)/\log\frac{2}{2-\varepsilon} + \delta$. We can substitute the parameters into the conclusion, and obtain that $\forall \alpha > 1, \lambda > 0, \delta > 0$, there exist $b = 1/\log\frac{2}{2-\varepsilon}$, $\varphi = \frac{\min\{\alpha-1,\lambda\}}{2\alpha+\delta-1+b(1+\alpha+\lambda)}$, and $\gamma = \frac{\alpha+1+b(1+\alpha+\lambda)}{2\alpha+\delta-1+b(1+\alpha+\lambda)}$, for any influence maximization algorithm based on $(\gamma, \varepsilon)$-almost submodular graphs, there exists a graph instance such that the approximation ratio cannot be higher than $N^{-\varphi}$, unless P=NP. Notice that $\varphi \geq \frac{\gamma}{3+3b}$ when we set $\alpha = \lambda + 1$ and $\lambda \geq 1$, and $\gamma$ ranges in $(0, 1)$, hence the theorem follows. $\qquad\square$

## B   Missing Proof of Theorem 3

The result follows the general framework on submodular functions laid out in the original work [8]. For convenience, we reproduce the proof specific to our Theorem 3 below.

*Proof.* Assuming that $S^*$ is the optimal set that maximizes the expectation of influenced nodes, $V_e$ are the set of $\ell$ nodes with non-submodular threshold function. let $S^*_{V_e} = S^* \cup V_e$. Apparently, $\sigma(S^*_{V_e}) \geq \sigma(S^*)$ since $\sigma(\cdot)$ is monotone. Since nodes in $V_e$ have non-submodular threshold functions, so we can not directly apply greedy scheme to find the seed set. But we can first add $V_e$ to the seed set and therefore the influence function $\sigma(S \cup V_e)$ is submodular when $S \subseteq V - V_e$. In this case, we obtain a greedy solution $S^g_{V_e,k}$ with adding extra $k$ nodes to original seed set $V_e$ with greedy scheme.

We assume that $S^*_{V_e} = V_e \cup \{s_1, s_2, \ldots, s_{k'}\}, k' \leq k, S_i = \{s_1, s_2, \ldots, s_i\}$, where $s_i \notin V_e$. Indeed, according to the proof of approximation ratio of $1 - 1/e$, for $i \leq k$, we have

$$
\begin{aligned}
&\sigma(S^*_{V_e}) \\
\leq\ &\sigma(S^*_{V_e} \cup S^g_{V_e,i}) \\
=\ &\sigma(S^g_{V_e,i}) + \sum_{j=1}^{k'}\left(\sigma(S^g_{V_e,i} \cup S_j) - \sigma(S^g_{V_e,i} \cup S_{j-1})\right) \\
\leq\ &\sigma(S^g_{V_e,i}) + \sum_{j=1}^{k'}\left(\sigma(S^g_{V_e,i} \cup \{s_j\}) - \sigma(S^g_{V_e,i})\right) \\
\leq\ &\sigma(S^g_{V_e,i}) + k \cdot \left(\sigma(S^g_{V_e,i+1}) - \sigma(S^g_{V_e,i})\right)
\end{aligned}
$$

The first line follows from monotonicity of $\sigma$, third line follows from the submodularity of $\sigma$, forth line holds because $S^g_{V_e,i+1}$ is built greedily from $S^g_{V_e,i}$ in order to maximize marginal benefit. Hence

$$
\begin{aligned}
\sigma(S^*_{V_e}) - \sigma(S^g_{V_e,i}) &\leq k \cdot \left(\sigma(S^g_{V_e,i+1}) - \sigma(S^g_{V_e,i})\right) \\
\sigma(S^*_{V_e}) - \sigma(S^g_{V_e,i+1}) &\leq \left(1 - \tfrac{1}{k}\right)\left(\sigma(S^*_{V_e}) - \sigma(S^g_{V_e,i})\right)
\end{aligned}
$$

When we apply greedy scheme for $k - \ell$ steps,

$$\sigma(S_{V_e}^*) - \sigma(S_{V_e, k-\ell}^g) \leq (1 - \tfrac{1}{k})^{k-\ell}(\sigma(S_{V_e}^*) - \sigma(V_e)) \leq (1 - \tfrac{1}{k})^{k-\ell}\sigma(S_{V_e}^*)$$

$$\sigma(S_{V_e, k-\ell}^g) \geq (1 - (1 - \frac{1}{k})^{k-\ell})\sigma(S_{V_e}^*) \geq (1 - e^{-\frac{k-\ell}{k}})\sigma(S^*)$$

Up till now we obtain that $\sigma(S_{V_e, k-\ell}^g) \geq (1 - e^{-\frac{k-\ell}{k}})\sigma(S^*)$. Hence we can first add all nodes without submodular threshold function to initial seed set and then apply greedy scheme to obtain a solution with theoretical approximation ratio. The theorem follows. $\qquad\square$