[Reviews · NeurIPS 2017]

Reviewer 1



The paper studies the influence maximization problem under the non-submodular model. The work differs from the traditional sub-modularity assumption. The problem is new and challenging. However, the paper has the following weakness: First, the \varepsilon-AS assumption is not convincing. I'd like to see data analysis to support the assumption. Second, the proofs are difficult to read. Intuitive explanations with examples are necessary to support the proofs.

Reviewer 2



This paper studies influence maximization in the general threshold diffusion model, specifically when node threshold functions are almost submodular (or epsilon-submodular). The key results are: 1) a proof of inapproximability for the influence maximization problem under this model when there are sufficiently many epsilon-submodular threshold nodes; 2) an efficient, simple and principled approximation algorithm for this problem when there is some fixed number of epsilon-submodular threshold nodes. This is a strong paper for the following reasons: 1- Influence maximization under the General Threshold model is a challenging and poorly studied problem; 2- The epsilon-submodularity condition is very reasonable because it is consistent with empirical evidence in real diffusion processes; 3- The hardness result and the approximate algorithms are intuitive and non-trivial. Barring minor typos and the like (see below for details), the paper is very well-written. In my understanding, the technical proofs are correct. The experimental evaluation is thorough and convincing, and the results are consistent with the theoretical guarantees. Minor comments (by line): 102: shouldn't the equation right-hand side have 1/(1-\epsilon) instead of (1-\epsilon)? 237: "linear" instead of "liner" 241: "satisfying" instead of "satisfies" 262: "PageRank" instead of "PagrRank" 297: "compared to 2.05" instead of "while 2.05" 298: "spread" instead of "spreads" 299: "The more $\epsilon$-AS nodes there are in the network, the more improvement is obtained"

Reviewer 3



Influence maximization is a #P hard problem to solve in general. A greedy algorithm is successful in giving a constant fraction approximate solution when the local influence functions (and by Mossel, Roche [8], also the global influence function) is submodular. This paper investigates the question of how well can one approximate influence if the local influence functions are not exactly submodular, but approximately submodular. Somewhat surprisingly, even when a only a sublinear fraction of the nodes are approximately submodular, the influence function cannot be approximated well in polynomial time. The authors establish this result by constructing a special example which "amplifies" the non-submodularity and gives rise to a non-approximable influence function. Furthermore the authors show that if only a constant number of nodes are approximately submodular, it is possible to obtain a constant fraction approximation, and the authors propose an algorithm to do the same. The paper considers a very interesting problem and has interesting theoretical results. The effectiveness of the algorithm in latter half of the paper is backed by simulations. I recommend this paper be accepted. Comments: -Line 24, please refer to authors by last names -Line 29, typo in LiveJournal